# The pregnant myometrium is epigenetically activated at contractility-driving gene loci prior to the onset of labor in mice

Virlana M. Shchuka[1☯¤]*, Luis E. Abatti[1☯¤], Huayun Hou[2,3], Nawrah Khader[1¤],
Anna Dorogin[4], Michael D. Wilson[2,3], Oksana Shynlova[4,5]*, Jennifer A. Mitchell[1¤]*

**1** Department of Cell and Systems Biology, University of Toronto, Toronto, Ontario, Canada, **2** Department of Molecular Genetics, University of Toronto, Toronto, Ontario, Canada, **3** Genetics and Genome Biology Program, SickKids Research Institute, Toronto, Ontario, Canada, **4** Lunenfeld Tanenbaum Research Institute, Sinai Health System, Toronto, Ontario, Canada, **5** Department of Obstetrics & Gynaecology, University of Toronto, Ontario, Canada

☯ These authors contributed equally to this work.
¤ Current address: Department of Cell and Systems Biology, University of Toronto, Toronto, Canada
* ja.mitchell@utoronto.ca (JAM); shynlova@lunenfeld.ca (OS); virlana.shchuka@mail.utoronto.ca (VMS)

**Data Availability Statement:** Sequencing data files were submitted to the Gene Expression Omnibus (GEO; https://www.ncbi.nlm.nih.gov/geo/) repository (GSE124295)

## Abstract

During gestation, uterine smooth muscle cells transition from a state of quiescence to one of contractility, but the molecular mechanisms underlying this transition at a genomic level are not well-known. To better understand these events, we evaluated the epigenetic landscape of the mouse myometrium during the pregnant, laboring, and postpartum stages. We generated gestational time point–specific enrichment profiles for histone H3 acetylation on lysine residue 27 (H3K27ac), histone H3 trimethylation of lysine residue 4 (H3K4me3), and RNA polymerase II (RNAPII) occupancy by chromatin immunoprecipitation with massively parallel sequencing (ChIP-seq), as well as gene expression profiles by total RNA-sequencing (RNA-seq). Our findings reveal that 533 genes, including known contractility-driving genes (Gap junction alpha 1 [*Gja1*], FBJ osteosarcoma oncogene [*Fos*], Fos-like antigen 2 [*Fosl2*], Oxytocin receptor [*Oxtr*], and Prostaglandin G/H synthase 2 (*Ptgs2*), for example), are up-regulated at day 19 during active labor because of an increase in transcription at gene bodies. Labor-associated promoters and putative intergenic enhancers, however, are epigenetically activated as early as day 15, by which point the majority of genome-wide H3K27ac or H3K4me3 peaks present in term laboring tissue is already established. Despite this early exhibited histone signature, increased noncoding enhancer RNA (eRNA) production at putative intergenic enhancers and recruitment of RNAPII to the gene bodies of labor-associated loci were detected only during labor. Our findings indicate that epigenetic activation of the myometrial genome precedes active labor by at least 4 days in the mouse model, suggesting that the myometrium is poised for rapid activation of contraction-associated genes in order to exit the state of quiescence.

**Funding:** This work was supported by the Canadian Institutes of Health Research (FRN 153198, held by JAM, OS, MDW, and VMS), the Canada Foundation for Innovation, and the Ontario Ministry of Research and Innovation (infrastructure grants held by JAM). Studentship funding was provided by the Natural Science and Engineering Research Council of Canada (CGS D held by VMS). The funders had no role in study design, data collection and analysis, decision to publish, or preparation of the manuscript.

**Competing interests:** The authors have declared that no competing interests exist.

**Abbreviations:** Akr1b7, Aldo-keto reductase family 1 member b7; AP-1, activator protein 1; Aqp8, Aquaporin 8; Cacna1e, Calcium channel voltage-dependent R type alpha 1E subunit; Calb2, Calbindin 2; Calml3, Calmodulin-like 3; Ceacam1, Carcinoembryonic antigen-related cell adhesion molecule 1; CEBP, CCAAT/enhancer-binding protein; ChIP-qPCR, ChIP-quantitative polymerase chain reaction; ChIP-seq, chromatin immunoprecipitation with massively parallel sequencing; Col11a1, Collagen type XI alpha 1; Col13a1, Collagen type XIII alpha 1; Col15a1, Collagen type XV alpha 1; Col26a1, Collagen type XXVI alpha 1; Col4a6, Collagen type IV alpha 6; CTCF, CCCTC-binding factor; Cxcl1, Chemokine C-X-C motif ligand 1; Cxcl5, Chemokine C-X-C motif ligand 5; eRNA, enhancer RNA; FOS, FBJ osteosarcoma oncogene; FOSL2, Fos-like antigen 2; GJA1, Gap junction alpha 1; GO, gene ontology; H3K4me1, histone H3 monomethylation of lysine residue 4; H3K4me3, H3 trimethylation of lysine residue 4; H3K9ac, acetylation of lysine residue 9 on histone H3; H3K14ac, acetylation of lysine residue 14 on histone H3; H3K27ac, H3 acetylation on lysine residue 27; H3K27me3, trimethylation of lysine residue 27 on H3; HDAC1, histone deacetylase 1; Hif3a, Hypoxia inducible factor 3 alpha subunit; HOMER, Hypergeometric Optimization of Motif Enrichment; IL-6, interleukin-6; iRNA-seq, intron RNA-seq; Jarid1A, Jumonji/ARID domain-containing protein 1a; JUN, Jun proto-oncogene; Kcng1, Potassium voltage-gated channel subfamily G member 1; Mchr1, Melanin-concentrating hormone receptor 1; miRNA, microRNA; miR-199a-3-p, microRNA precursor 199a-3-p; miR-214, microRNA precursor 214; Mmp7, Matrix metalloproteinase 7; Mmp9, Matrix metalloproteinase 9; Mmp11, Matrix metalloproteinase 11; Mmp12, Matrix metalloproteinase 12; Oxtr, Oxytocin receptor; PRA, progesterone receptor A; Ptgs2, Prostaglandin G/H Synthase 2; RELA (NFkB-p65), Nuclear factor-kappa-B p65 subunit; RNAPII, RNA

## Introduction

Over the course of gestation, the myometrium transitions from a state of quiescence during pregnancy to one of contractile activity during labor in response to both hormonal and mechanical signals. Concomitant changes in gene expression that accompany this transition are thought to be a driving force for the initiation of labor [1,2]; however, little is known about the molecular mechanisms underlying these changes. Across developmental contexts, the chromatin landscape is thought to maintain a cell's identity. Dynamic chromatin states can distinguish one cell type from another and account for differences between cells of the same type under disparate environmental conditions [3]. Across cell types, transcription start sites (TSSs) of actively transcribed genes are marked by histone H3 trimethylation of lysine residue 4 (H3K4me3) and H3 acetylation on lysine residue 27 (H3K27ac), and increased gene expression levels are correlated with the presence of both markers at gene TSSs [4–8]. Additionally, transcriptional enhancers, which can be located at kilobase-to-megabase-sized distances from the genes they regulate, contain a prominent signature consisting of H3K27ac [3,4,9–11] and noncoding enhancer RNAs (eRNAs), both of which can be used as a means of identifying regions with tissue-specific enhancer activity [12–14]. Finally, the presence of histone modifications typically associated with active genes and the subsequent recruitment of RNA polymerase II (RNAPII) to gene promoters allow for transcription initiation and transition to elongation, thereby up-regulating gene expression [15]. Where, how, and at what point these events occur in the myometrial genome during gestation are the inquiries guiding this study.

We know that uterine contractions are enabled when myometrial muscle cells act en masse to generate a series of synchronous movements, actions that require the coupling of cells through the presence of intercellular bridges, or gap junctions. Among the proteins mediating junction formation as term approaches, Gap junction alpha 1 (GJA1, also known as CX43) is most prominently up-regulated [16]. Selective reduction of GJA1 production in the uterine smooth muscle cells of two different mouse models has been shown to significantly prolong the quiescent state during pregnancy and thereby delay the onset of labor [17,18]. Reporter expression downstream of a synthetic *Gja1* promoter is increased by coexpression of constructs encoding members of the activator protein 1 (AP-1) transcription factor FBJ osteosarcoma oncogene (FOS) and Jun proto-oncogene (JUN) subfamilies [19–21]. In rodent and human labor, JUN protein levels remain fairly constant in the myometrium throughout gestation, whereas increased levels of FOS and Fos-like antigen 2 (FOSL2) proteins are observed during labor within the nuclei of myometrial cells. Despite the presence of several JUN subfamily members in the uterine smooth muscle during quiescent stages of pregnancy, their displayed ability to act as homodimerized activators of *Gja1* promoter-driven transcription in reporter assays is more limited compared with that of heterodimers composed of FOS and JUN subfamily members [22,23]. It is therefore likely that JUN protein members may have a role in maintaining myometrial gene expression during pregnancy but require heterodimerization with a FOS subfamily partner to activate genes required for the onset of labor.

Despite extensive in vitro studies correlating FOS:JUN activity with *Gja1* promoter activation and consequent labor initiation, little is known about the active chromatin landscape on a genome-wide scale in the myometrium as uterine smooth muscle cells exit the quiescent phase and enter the laboring state. We address this gap in the literature by investigating the epigenetic and transcriptomic changes that take place in the nucleus during this cellular transition. Using total RNA-sequencing (RNA-seq) methods, we observed an increase in primary transcript levels for the majority of genes that display increased expression during labor, suggesting that the initiation of contractility involves substantial modulation of gene transcription. Despite these temporally dependent differences in transcription output, the myometrial

polymerase II; RNA-seq, RNA-sequencing; RPM, reads per million; Spock2, Sparc/osteonectin, cwcv and kazal-like domains proteoglycan 2; TCF3, Transcription factor 3; Thbs1, Thrombospondin 1; TSS, transcription start site; Vcam1, Vascular adhesion molecule 1; ZEB1, Zinc finger E-box-binding homeobox protein 1; ZEB2, Zinc finger E-box-binding homeobox protein 2.

genome does not undergo a corresponding acquisition of euchromatin-associated histone marks. Instead, we determined that H3K27ac and H3K4me3 modifications are present at labor–up-regulated gene promoters during the uterine quiescent stage, several days prior to the onset of labor. Although gene promoters are premarked with these histone modifications, we identified increased RNAPII enrichment at promoters and across gene bodies and increased expression of eRNAs in noncoding regions surrounding labor-associated genes during active labor. Furthermore, we found that intergenic regions exhibiting H3K27ac peaks and labor–up-regulated eRNA expression displayed an enrichment of AP-1 transcription factor motifs, thereby implicating FOS and JUN proteins in the distal regulation of gene transcription changes at labor onset. These observations collectively suggest that the murine myometrium undergoes a cascade of epigenetic events that begins well in advance, and continues to the commencement, of labor at term.

## Results

### Up-regulation of labor-associated genes involves a transcriptional mechanism

To establish a comprehensive profile of pregnant and laboring myometrial transcriptomes, we conducted total strand-specific RNA-seq on RNA isolated from the myometrium of pregnant C57BL/6 mice at gestational day 15 or day 19 while in active labor ($n$ = 5 each, Fig 1A). Based on the RNA-seq data, we observed clustering of the same samples within each time point of collection, as expected (S1 Fig and S1 Data). Differential gene expression analysis based on exon read counts (S1 Table) revealed that a total of 956 genes showed gestational time point–varying expression levels (Fig 1B, fold change cutoff of 4, $p < 0.01$). Hierarchical clustering analysis of these genes confirmed similar expression trends from mice of the same gestational age (Fig 1C and S2 Data) while gene ontology (GO) term analyses highlighted the involvement of down-regulated and up-regulated genes at term in myometrial relaxation and contraction pathways, respectively (S2 and S3 Tables). In all, 578 genes exhibited a significant increase in expression during active labor compared with day 15. Apart from up-regulation of *Fos* (Fig 1D), these genes included (but were not limited to) prominent labor-associated players *Gja1*, Prostaglandin G/H synthase 2 (*Ptgs2*), and Oxytocin receptor (*Oxtr*), as well as matrix metalloproteinases (Matrix metalloproteinases 7 [*Mmp7*], 11 [*Mmp11*], and 12 [*Mmp12*]), signaling proteins (Chemokine C-X-C motif ligands 1 [*Cxcl1*], and 5 [*Cxcl5*]), and adhesion molecules and proteins (Vascular adhesion molecule 1 [*Vcam1*], Thrombospondin 1 [*Thbs1*], Carcino-embryonic antigen-related cell adhesion molecule 1 [*Ceacam1*]) known to exhibit elevated levels at term. Conversely, 378 genes were found to be significantly down-regulated during active labor compared with day 15, including (but not limited to) genes encoding proteins responsible for cell–extracellular matrix interactions (Collagens type IV alpha 6 [*Col4a6*], type XI alpha 1 [*Col11a1*], type XIII alpha 1 [*Col13a1*], type XV alpha 1 [*Col15a1*], type XXVI alpha 1 [*Col26a1*], and Sparc/osteonectin, cwcv and kazal-like domains proteoglycan 2 [*Spock2*]), proteins involved in calcium signaling (Melanin-concentrating hormone receptor 1 [*Mchr1*], Cal-modulin-like 3 [*Calml3*], Calbindin 2 [*Calb2*]), proteins regulating myometrium response to low oxygen tension and resistance to oxidative stress (Hypoxia inducible factor 3 alpha subunit [*Hif3a*], Aldo-keto reductase family 1 member b7 [*Akr1b7*]), and voltage-dependent calcium, potassium, and water channels (Calcium channel voltage-dependent R type alpha 1E subunit [*Cacna1e*], Potassium voltage-gated channel subfamily G member 1 [*Kcng1*], and Aquaporin 8 [*Aqp8*], respectively).

Differential exonic RNA profiles, however, do not in and of themselves reflect a regulatory mechanism change at the level of transcription for those genes. Mediation of gene expression

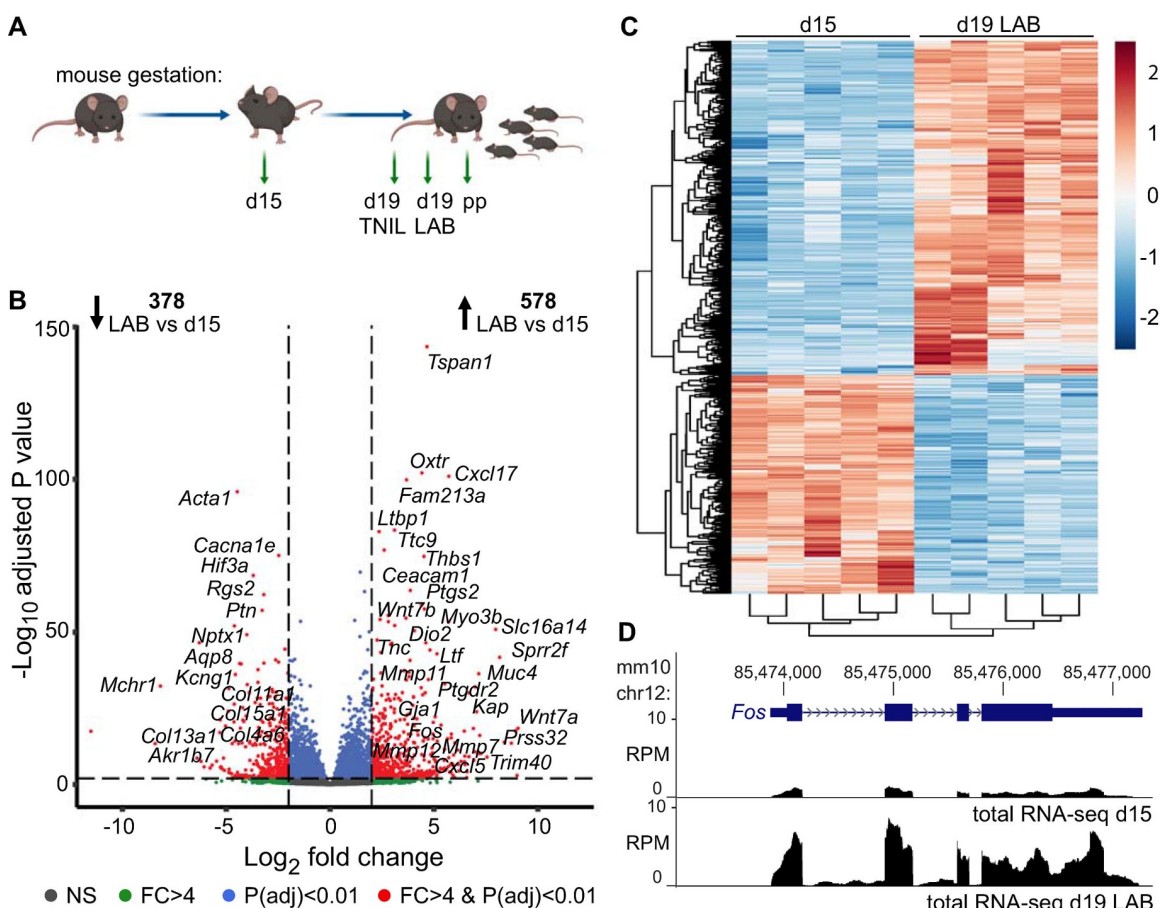

**Fig 1. Quiescent and term laboring myometrial transcriptomes exhibit differential expression profiles.** (A) Gestational schematic outlining days and time points at which myometrial tissues were collected for transcriptome- or genome-wide sequencing analyses. Collection days include gestational d15, d19 TNIL, d19 LAB, and pp. (B) RNA-seq volcano plot highlighting transcriptional status of genes exhibiting differential expression levels between d15 and d19 LAB myometrial tissues. Data associated with this figure can be found in S1 Table. (C) Hierarchical clustering of gene groups based on RNA expression changes between d15 and d19 LAB samples (*n* = 5 per gestational day). Data associated with this figure can be found in S2 Data. (D) Total RNA-seq reads (RPM) at the labor-associated *Fos* gene locus for d15 and d19 LAB samples mapped to the mm10 mouse genome assembly. chr12, chromosome 12; d, day; FC, fold-change; LAB, active labor; NS, nonsignificant; pp, postpartum; RNA-seq, RNA-sequencing; RPM, reads per million; TNIL, term-not-in-labor.

regulation can take place at multiple stages within a gene's expression pathway prior to formation of a final translated and functional protein product. Apart from varying of levels of primary transcript generation in the nucleus, these mechanisms can include nuclear retention of processed mRNA [24,25], alternative splicing [26,27], and increased mRNA stability [28,29]. Furthermore, multiple gene activation and repression mechanisms can compete to determine the expression outcome of a particular gene, as demonstrated by select biological contexts in which there is an imperfect correlation between the levels of a gene's transcribed and its translated products [30].

Because sequencing of total RNA provides a transcriptome-wide profile of reads spanning both exons and introns, both spliced and unspliced RNA species can be detected [31,32]. If critical labor-driving genes were significantly up-regulated at the level of active transcription, we would expect to identify a substantial increase in reads corresponding to gene introns as well as those corresponding to their exons. Conversely, if activated genes were regulated

exclusively or predominantly by posttranscriptional mechanisms such as RNA stability, we would expect that those genes' corresponding intron reads would remain relatively constant throughout gestation. Given that a prior study has posited that mRNA stability may act as a critical player in regulation of *Gja1* in particular [33], we sought to definitively ascertain whether increased contractility-associated gene expression during labor involved any regulatory input from transcriptional mechanisms. Upon inspection of the *Fos* and *Gja1* loci, we noted striking labor-specific intronic RNA enrichment profiles at both genes (Fig 2A). Subsequently, we conducted a genome-wide intron reads–concentrated gene expression analysis (intron RNA-seq [iRNA-seq]) [31], which uncovered multiple genes that displayed increased primary transcript generation at term during active labor relative to day 15 (Fig 2B and S4 Table). In fact, the majority of genes (55%) up-regulated at labor on the basis of increased exon read accumulation also displayed a significant increase in intron reads (Fig 2C). Using exon–

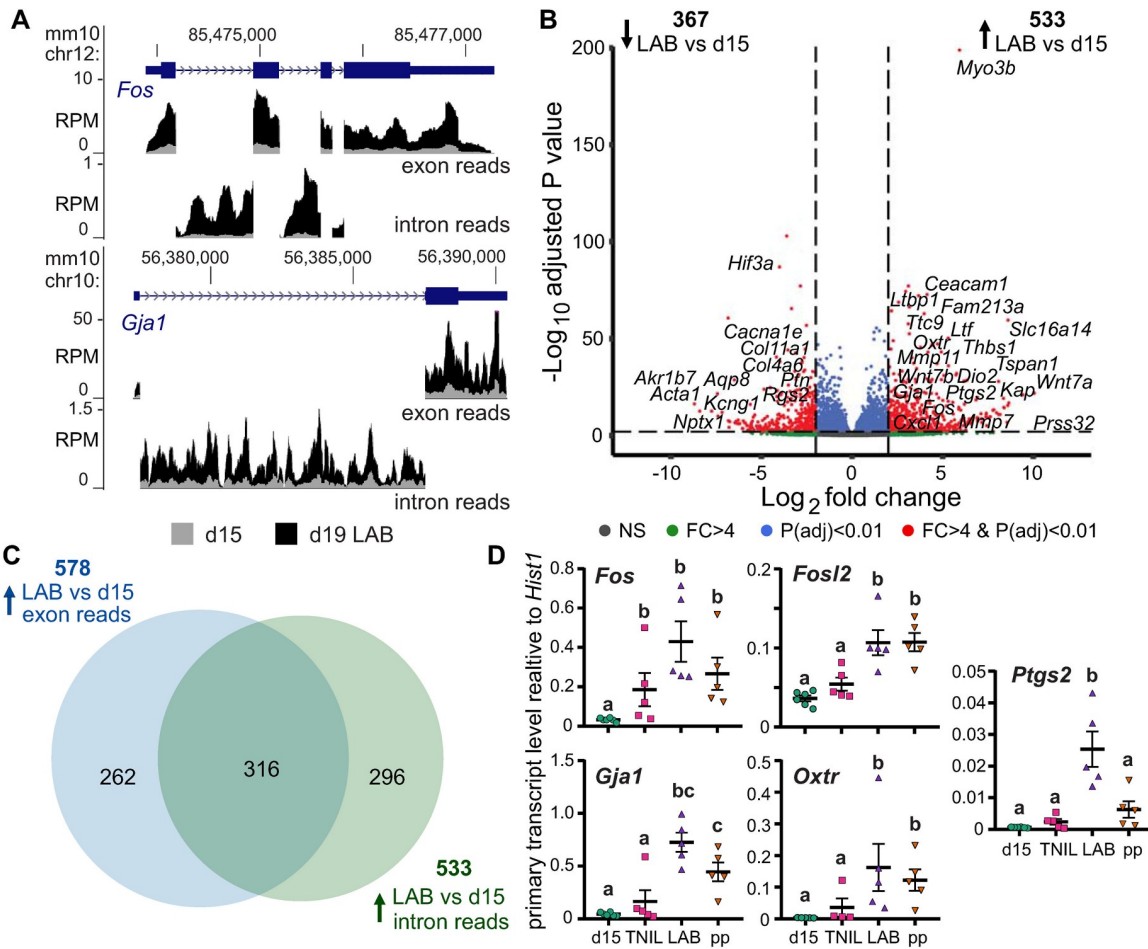

**Fig 2. Up-regulation of labor-associated genes involves an increase in their primary transcript levels.** (A) Exon- and intron-specific RNA-seq reads (RPM) at the labor-associated *Fos* gene locus from d15 and d19 LAB samples, mapped to the mm10 mouse genome assembly. (B) iRNA-seq volcano plot highlighting expression changes based on intron reads between d15 and d19 LAB samples. (C) Venn diagram displaying the number of genes significantly up-regulated during labor relative to d15 that show enrichment in exon and intron reads (region of overlap) or enrichment in either exon or intron reads (regions of nonoverlap). (D) Confirmation of laboring time point–specific up-regulation of primary transcript expression of contractility-promoting genes by RT-qPCR. Groups labeled with different letters show significant difference, with $p < 0.05$. Data associated with this figure can be found in S3 Data. chr12, chromosome 12; d, day; FC, fold-change; iRNA-seq, intron RNA-seq; LAB, active labor; NS, nonsignificant; P(adj), adjusted p-value; RNA-seq, RNA-sequencing; RPM, reads per million; RT-qPCR, reverse transcriptase-quantitative polymerase chain reaction; TNIL, term-not-in-labor.

intron junction-spanning primers, we confirmed a significant increase in primary transcript levels of well-known labor-associated genes—*Gja1*, *Fos*, *Fosl2*, *Oxtr*, and *Ptgs2*—in active labor relative to day 15 and, with the exception of *Fos*, to day 19 term-not-in-labor (Fig 2D and S3 Data). These results demonstrate that genes promoting the contractile state in the myometrium at term act because of a rapid gestational time point–specific increase in primary transcript levels, which suggests that substantial transcriptional activity occurs in myometrial nuclei during labor.

## Histone marks associated with gene activation are enriched at labor-associated gene promoters well in advance of labor onset

Having determined that the majority of genes exhibiting increased expression levels during labor were associated with increased primary transcript generation, we next investigated the active chromatin landscape surrounding these genes. After optimizing the protocol for myometrial tissue and confirming target enrichment at control regions (S2 Fig and S4 Data), we conducted chromatin immunoprecipitation with massively parallel sequencing (ChIP-seq). We targeted H3K4me3 and H3K27ac enrichment events on a genome-wide scale at day 15, day 19 term-not-in-labor, day 19 active labor, and postpartum, and found gestational time point sample replicates were highly correlated, as expected ($R^2 > 0.79$, S5 Table). When we compared the histone profiles at all promoters to the corresponding genes' transcriptional status within a particular time point, we observed that both H3K4me3 and H3K27ac are enriched at the promoters of genes highly expressed on the designated gestational day (S3 Fig and S5 Data). However, an initial examination of the *Fos* locus unexpectedly revealed an enrichment of both active chromatin markers at the *Fos* promoter across all four time points (Fig 3A); furthermore, the same trend was observed for labor-associated genes *Fosl2*, *Gja1*, *Oxtr*, and *Ptgs2* (S4 Fig, histone panels). We next sought to establish the active histone profile across all gene promoters at the designated gestational stages (S6/S7 Tables). Even more surprisingly, we found a similar genome-wide active histone marker enrichment pattern across gene promoters (+/− 2 kb of TSSs) in all four time points (Fig 3B). A K-means clustering analysis further revealed groups of promoters with differing H3K27ac and H3K4me3 enrichment levels but failed to identify groups of genes displaying gestational day–associated changes in these histone modifications' profiles (S5 Fig). When we investigated the levels of these modifications at promoters of exclusively those genes that displayed a significant increase in transcription (based on an increase in intron reads) during active labor, we observed only a moderate increase in accumulation of both markers from day 15 to term (Fig 3C and 3D and S6, S7 and S8 Datas). Our analyses therefore indicate that H3K27ac and H3K4me3 enrichment premarks the promoters of labor–up-regulated genes as early as day 15 of gestation.

## Loci of labor-associated genes acquire RNAPII gene body occupancy and eRNA enrichment events closer to term

Given that labor-associated gene loci exhibited strikingly similar active histone mark enrichment profiles across all four gestational time points, we next sought to establish whether concomitant binding of RNAPII, an event required for transcription initiation, occurred at contractility-driving gene promoters and bodies as early in the gestational time course. We conducted ChIP-seq to identify RNAPII enrichment events (targeting serine 5 phosphorylated RNA Polymerase II Subunit A) at gestational days 15 and 19 (during active labor) and found RNAPII-enriched broad peaks in both stages (S8 Table). Again, our correlation analysis confirmed the clustering of gestational time point replicates (S6 Fig and S9 Data) while an examination of RNAPII enrichment values alongside our RNA-seq data revealed that RNAPII has a more prominent

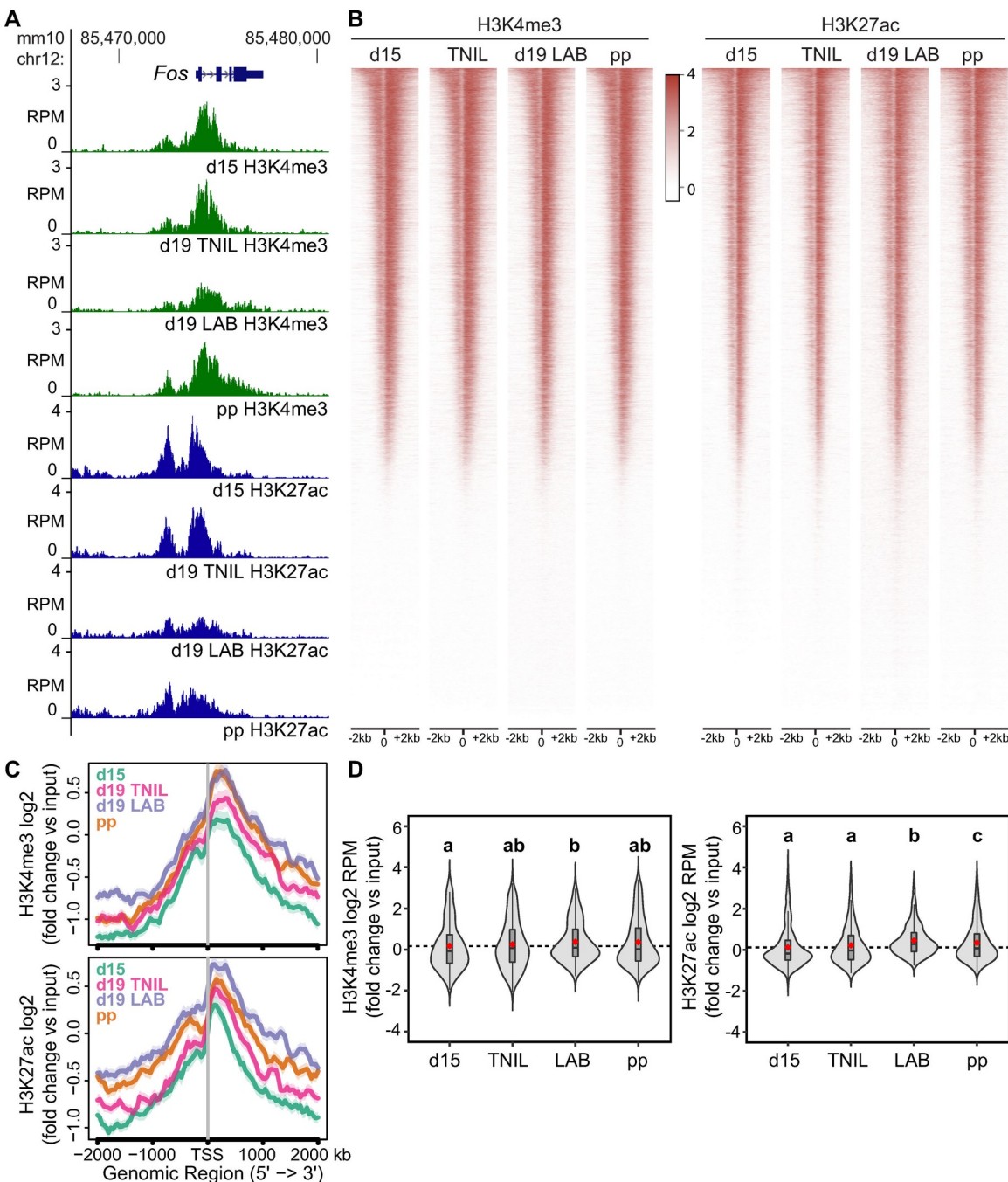

**Fig 3. Gene expression activation–associated histone marks are enriched at promoters of contractility-driving genes in advance of labor onset.** (A) Anti-H3K4me3 and anti-H3K27ac ChIP-seq reads (RPM) mapped at promoter of labor-associated gene *Fos* in d15, d19 TNIL, d19 LAB, and pp samples. (B) Genome-wide enrichment of H3K4me3 and H3K27ac at gene promoters throughout the genome. Signal +/− 2 kb of TSSs is displayed, with genes ordered at each indicated time point according to decreasing enrichment (red → white) profile in d15 samples. (C) Plots exhibiting log$_2$-fold H3K4me3 or H3K27ac signal, normalized to input, at promoters of genes whose expression is enriched in laboring samples relative to d15 samples based on intron reads. Data associated with this figure can be found in S6 Data. (D) Violin plots displaying log$_2$-fold H3K4me3 or H3K27ac signal (RPM), normalized to input, at promoters of genes whose intron read–based expression is enriched in laboring samples relative to d15 samples. Groups labeled with different letters show significant difference, with $p < 0.05$. Data associated with this figure can be found in S7 and S8 Datas. ChIP-seq, chromatin immunoprecipitation with massively parallel sequencing; chr12, chromosome 12; d, day; *Fos*, FBJ osteosarcoma oncogene; H3K4me3, H3 trimethylation of lysine residue 4; H3K27ac, H3 acetylation on lysine residue 27; LAB, active labor; pp, postpartum; RPM, reads per million; TNIL, term-not-in-labor; TSSs, transcription start site.

binding profile at highly expressed genes (S7 Fig and S5 Data and S9 Table). Contrary to the active histone marker trend we observed, labor-driving gene promoters exhibited differential RNAPII binding profiles at either gestational stage. We found that promoter and gene body polymerase occupancy at genes with substantially higher expression levels during labor was significantly higher at term relative to day 15 (Fig 4A and 4B and S6 and S10 Datas); such genes include, but are not limited to, *Fos*, *Gja1*, *Ptgs2*, and *Fosl2* (Figs 4C and S4). This result correlated with our observations of these genes' gestational time point–specific primary transcript levels and further supported the notion that transcriptional mechanisms underlie the rapid and prompt gene expression up-regulation events that underlie the myometrial state transition toward contractility.

Among the genes significantly up-regulated during labor, *Fos* exhibited increased RNAPII occupancy across its gene body relative to day 15, as we expected. When we further expanded our view outside the gene body, we observed a genomic region that does not encode a gene but includes both a prominent RNAPII peak (marked by red bar within dashed lines in Fig 4C) as well as an intergenic H3K27ac enrichment event 12 kb downstream of the *Fos* gene. As was the case with the H3K27ac signal at the *Fos* promoter, an intergenic H3K27ac peak was identified in both day 15 as well as day 19 labor samples; however, RNAPII association at this region was more pronounced and only identified as a peak in the labor context (Fig 4C). Because intergenic regions containing RNAPII and H3K27ac peaks have been noted to occur at active enhancer regions [6,12,34], we examined the loci of other labor-associated genes to see whether they contained intergenic regions that similarly exhibit this epigenetic profile. We found that, although we focused our earlier analyses on H3K27ac signal enrichment at promoter regions, 43% of the identified H3K27ac peaks in the laboring samples are located at distances greater than 2 kb from a gene TSS (Fig 4D and S10 Table). As was the case with H3K27ac peaks at promoters throughout the genome, intergenic H3K27ac peaks were mostly invariant across our tested gestational time points, with only 11/5,041 (0.2%) of the intergenic H3K27ac peaks displaying a significant increase in H3K27ac signal enrichment in labor samples compared with day 15 samples (S11 Table). However, despite the presence of H3K27ac enrichment at intergenic regions across all four tested time points, many gene loci contain H3K27ac-modified regions with associated differentially transcribed eRNAs (S12 Table). Furthermore, we observed that several of these regions contain H3K27ac peaks and display a significant increase in eRNA expression levels during labor (Fig 4D). Therefore, although most H3K27ac peaks across the myometrial genome are present both at day 15 and at term, several regions containing those peaks transcribe significantly higher amounts of eRNA at term.

To identify the transcription factor motifs that could underlie the changes in gene expression that occur during labor, we conducted motif enrichment analyses using Hypergeometric Optimization of Motif Enrichment (HOMER) [35]. We uncovered a significant enrichment of AP-1 motifs (TGACTCA) in labor-associated eRNA–up-regulated intergenic regions (Fig 4E), implicating this family of transcription factors in a modulatory role with regard to enhancer activity at labor onset. We made a similar observation at promoters of genes displaying increased primary transcript abundance during labor (Figs 4F, S8 and S6 Data). Additionally, apart from AP-1 motifs, the promoters of these genes contain an enrichment of several other motifs: Transcription factor 3 (TCF3), an E-protein transcription factor that has been shown to assist coactivator proteins in the induction of gene transcription events in other cell contexts [36]; CCCTC-binding factor (CTCF), a zinc finger protein prominently known to bind promoter and enhancer regulatory elements [37]; and RELA or NFkB-p65 (Nuclear factor-kappa-beta p65 subunit), a member of the labor-associated NFkB-p65/IL-6 inflammatory pathway [38,39] that has also been affiliated with inducing transcription at the *Oxtr* promoter [40]. Taken together, these results lead us to propose that the controlled expression of labor-

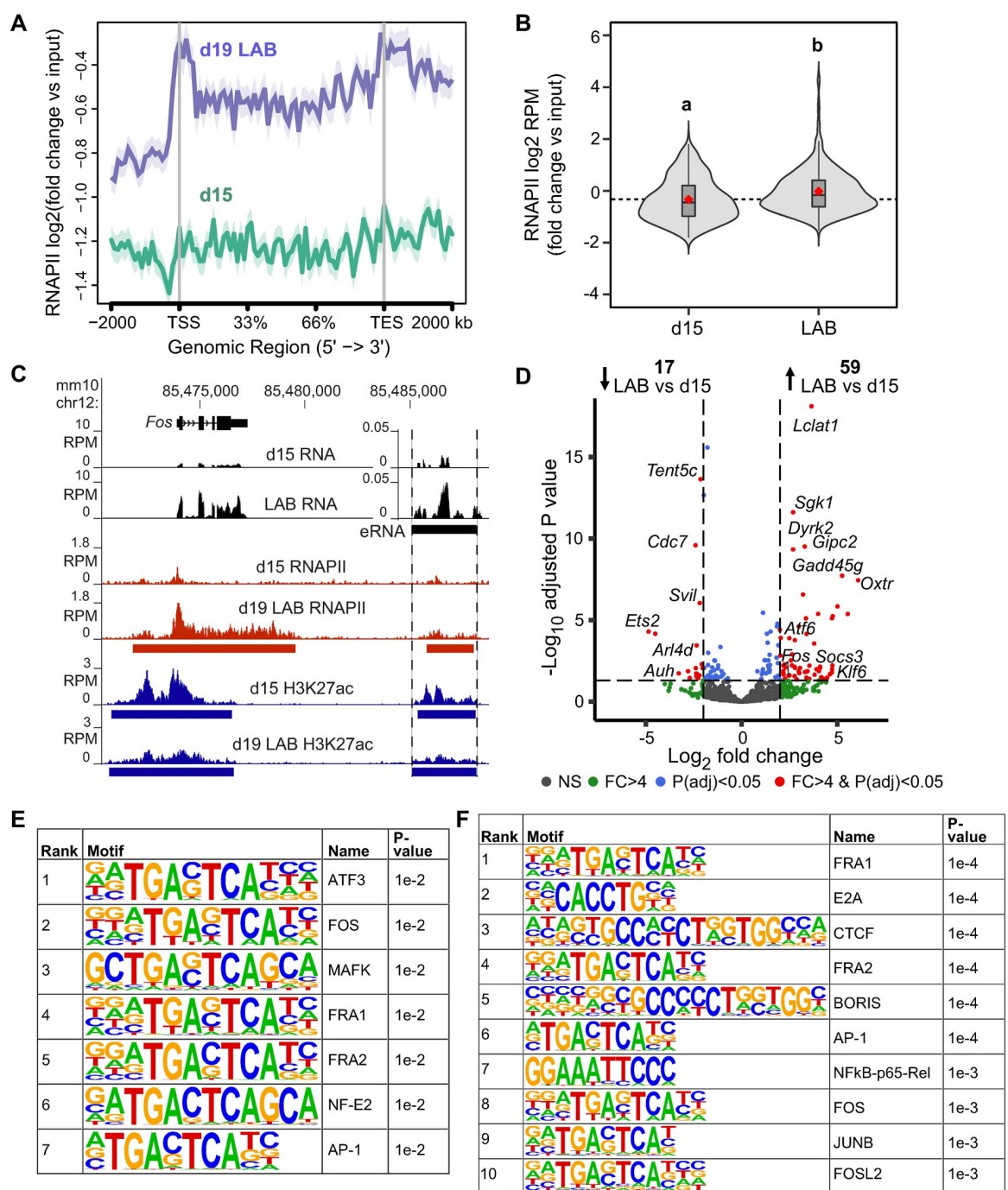

**Fig 4. Active labor is associated with recruitment of RNAPII to labor-driving genes and eRNA expression at gene-adjacent intergenic regions with H3K27ac peaks.** (A) Metagene plot exhibiting log$_2$-fold RNAPII signal, normalized to input, at bodies of genes whose intron read–based expression levels are enriched in d19 LAB samples relative to d15 samples. Data associated with this figure can be found in S6 Data. (B) Violin plots displaying log$_2$-fold RNAPII signal (RPM), normalized to input, at promoters of genes whose intron read–based expression levels are enriched in d19 LAB samples relative to d15 samples. Groups labeled with different letters show significant difference, with $p < 0.05$. Data associated with this figure can be found in S10 Data. (C) RNA-seq, RNAPII, and H3K27ac ChIP-seq reads (RPM) mapped at the *Fos* locus in d15 or d19 LAB samples. Regions containing RNAPII (red bars) or H3K27ac peaks (blue bars) indicated in d15 and d19 LAB samples. RNA-seq profile at labor–up-regulated eRNA-exhibiting region (situated between the dashed lines) downstream of *Fos* is presented at a different scale compared with the rest of the locus. (D) eRNA volcano plot highlighting intergenic H3K27ac peaks in genomic regions that exhibit significant differences in eRNA levels between d15 and d19 LAB samples. (E) All enriched transcription factor motifs identified at genomic regions that feature intergenic H3K27ac peaks and exhibit a significant increase in eRNA levels in d19 LAB samples compared with d15 samples. (F) The top 10 enriched motifs identified at promoters for

genes that exhibit a significant increase in intronic RNA levels for d19 laboring samples compared with d15 samples. ChIP-seq, chromatin immunoprecipitation with massively parallel sequencing; chr12, chromosome 12; d, day; eRNA, enhancer RNA; FC, fold-change; *Fos*, FBJ osteosarcoma oncogene; H3K27ac, H3 acetylation on lysine residue 27; LAB, active labor; NS, nonsignificant; RNAPII, RNA polymerase II; RNA-seq, RNA-sequencing; RPM, reads per million; P(adj), adjusted p-value; TES, transcription end site; TSS, transcription start site.

associated genes is driven by transcriptional regulation mechanisms, despite the apparent epigenetic activation of labor-associated loci well in advance of labor onset.

## Discussion

Drawing upon both our and previously established data, we offer a biological model wherein the myometrium's preparation for labor at a genomic level begins well in advance of term (Fig 5). We propose that the presence of H3K27ac and H3K4me3 marks at labor-associated gene promoters and putative intergenic enhancers in the precontractile mouse myometrium as early as gestational day 15 epigenetically activates these regulatory regions, even at a stage of pregnancy when the expression of many labor-associated genes is comparatively low. During pregnancy, promoters and intergenic regions may be bound by homodimerized members of the JUN transcription factor family, proteins that are already significantly expressed at this stage. In time, the combined action of hormonal and mechanical signals induces the process of labor, the commencement of which involves novel transcriptomic events: as we observed, increased contractility-driving gene and eRNA transcription occurs within labor-associated loci from promoter and noncoding regions, respectively. These proximal and distal regulatory regions both contain prominent H3K27ac peaks and are enriched for AP-1 sequence motifs. We suggest that these motifs, if resident in regions of open chromatin and formerly bound by JUN proteins, may allow for a laboring stage–specific replacement of homodimers with FOS: JUN family member heterodimers, as well as phosphorylated RELA (both of which have been implicated in labor onset). Such labor-associated transcription factor binding actions could enable what our data indicate is a gestational time point–specific recruitment of RNAPII and consequent primary transcript production. Collectively, these intranuclear events can form the regulatory mechanistic basis of the myometrial organ's transition from a quiescent to a contractile state.

To our knowledge, this study is the first to demonstrate that key contractility-promoting genes in the myometrium are up-regulated at term at least in part because of a significant increase in primary transcript abundance. This finding does not preclude the notion that regulation mechanisms act at multiple stages in the expression pathways of labor-associated genes to mediate their expression output. Regulation of *Ptgs2*, for instance, involves microRNA (miRNA)-mediated repressive mechanisms during pregnancy that are halted by reduced expression of microRNA precursors 199a-3-p (miR-199a-3-p) and 214 (miR-214) as gestation progresses toward term [41,42]. However, our findings reveal that the onset of labor depends on substantial term-restricted transcriptional activity of *Ptgs2*. Whereas our work investigates these regulatory mechanisms on a genome-wide scale, much of the prior scholarship on labor-associated gene activation events from a single-gene perspective has supported this notion. In vitro studies have confirmed the regulatory role of select nuclear factor binding sites in critical genes' promoters: for instance, mutation of an AP-1 factor binding site in the synthetic *Gja1* promoter has been shown to inhibit AP-1 factor–mediated reporter gene expression [21]. Similarly, Khanjani and colleagues have demonstrated that a 20-base-pair-long genomic segment upstream of the human *Oxtr* promoter is required for reporter expression, which is mediated by nuclear factors CCAAT/enhancer-binding protein (CEBP) and RELA [40]. Furthermore,

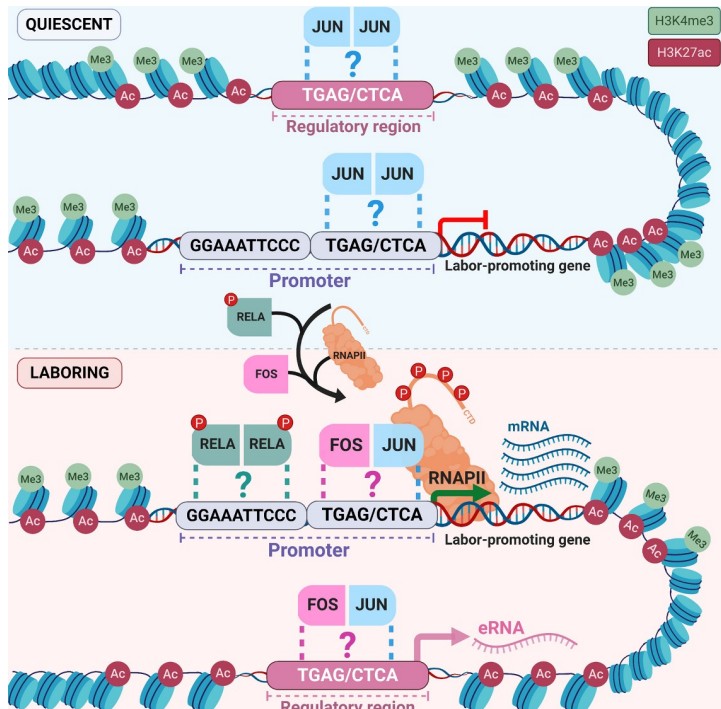

**Fig 5. Model of epigenetic priming and transcriptional regulation mechanisms that initiate increased gene expression during labor.** Display of the chromatin landscape around typical labor–up-regulated genes at quiescent (above) and term laboring (below) stages of gestation. Quiescent: during pregnancy, H3K27ac and H3K4me3—histone markers typically associated with active genes—are already present at labor-associated gene promoters and putative intergenic enhancers, thereby priming the epigenome for contractility-driving transcriptional events in advance of term. The aforementioned regions also contain an enrichment of AP-1 transcription factor motifs that may allow for potential binding of homodimerized JUN proteins, which are already known to be expressed in the quiescent stage. Laboring: conversely, the progression of gestation toward term induces transcription events at premarked intergenic regions and results in eRNA accumulation within key gene loci. Putative regulatory region–bound JUN proteins may be replaced by FOS:JUN proteins alongside phosphorylated RELA. This hypothetical switch in dimerized transcription factor binding events may underlie the labor-specific recruitment of RNAPII to labor-associated gene promoters that we have observed and, consequently, enhance labor-specific transcription through the bodies of these genes. Molecular events depicted in this figure that have not been experimentally validated in our or previous studies are marked by "?". Ac, acetylation; AP-1, activator protein 1; CTD, C-terminal domain; FOS, FBJ osteosarcoma oncogene; H3K4me3, H3 trimethylation of lysine residue 4; H3K27ac, H3 acetylation on lysine residue 27; JUN, Jun proto-oncogene; Me3, trimethylation; P, phosphorylation site; RELA, Nuclear factor kappa beta p65 subunit; RNAPII, RNA polymerase II.

ChIP-quantitative polymerase chain reaction (ChIP-qPCR) experiments conducted in several studies revealed gestational time point–specific binding events that correspond to differential labor-associated gene expression outputs. Renthal and colleagues, for instance, have shown that an intracellular abundance of transcriptional repressors Zinc finger E-box-binding homeobox proteins 1 (ZEB1) and 2 (ZEB2) inversely correlates with Oxtr and Gja1 mRNA levels in myometrial cells; furthermore, prominent endogenous binding of ZEB1 and ZEB2 at *Gja1* and *Oxtr* promoters during pregnancy dramatically reduces by term [43]. Finally, increased expression of progesterone receptor A (PRA), a protein critical for contractility in the human myometrial laboring context, is thought to occur because of reduced Histone deacetylase 1 (HDAC1) binding and Jumonji/ARID domain-containing protein 1a (JARID1A) histone demethylase enrichment at the PRA promoter [44–46]. These studies suggest that enrichment of activation-prompting histone acetylation and methylation markers at promoters of labor-associated genes guide the transition of the myometrium to a contractile state. Indeed, transcription studies that describe the transition of a particular cell type to another

state upon subjection to different environmental conditions typically show evidence of clear histone mark turnover at critical transition-guiding genes [9,47,48]. Contrary to this model, we observed similar trends in H3K27ac and H3K4me3 occupancy across all four tested time points in mouse myometrial samples, with only subtle acquisition of activating histone marks at promoters of significantly labor–up-regulated genes, instead of a clear marker loss or gain according to myometrium state. Furthermore, intergenic regions of labor-associated genes contain H3K27ac peaks that are called as early as day 15, suggesting that not only promoters but even putative enhancers required for labor initiation may be established during the quiescent phase of pregnancy.

The precise time point at which labor-driving regulatory regions begin to exhibit the activating histone profile we observed, however, remains to be determined. Though taking place in the final quarter of the murine gestational period, the gestational day 15 time point nevertheless corresponds to a myometrial cellular state of dormancy. This noncontractile phenotype is accompanied by a substantially disparate transcriptome from its contractile counterpart, as both our work here and prior studies [1,49,50] have demonstrated. In light of these molecular-level expression and tissue-level phenotypic differences, we initially hypothesized that the promoters of labor–up-regulated genes would not have acquired activating histone marks on day 15. Our finding to the contrary raises the possibility that labor-associated regions are enriched by H3K27ac and H3K4me3 modifications still earlier than this day. Whether these activating marks are first acquired at a time point during early or midgestation or, perhaps more intriguingly, whether such epigenetic profiles exist in the nonpregnant state and persist throughout gestation requires further investigation. Comparing the profiles of labor-driving gene loci in pregnant mouse myometrium at gestational time points prior to day 15 with those of myometrium in nonpregnant female mice would definitively ascertain what window of time, if any, precedes the establishment of activating histone mark deposition at critical regulatory elements.

Our study's H3K27ac and H3K4me3 enrichment data at both promoters and intergenic regions provide a critical first look at the state of chromatin in myometrial cells at different gestational stages. Future consideration of other epigenetic markers can more discriminately trace significant chromatin-altering events that occur during pregnancy. For instance, apart from H3K4me3, gene promoters undergoing a readying process for strong expression induction are characterized by pronounced enrichment of additional modifications, including acetylation of lysine residues 9 and 14 on histone H3 (active markers H3K9ac and H3K14ac) [51,52]. Furthermore, "bivalent promoters," or promoters simultaneously laden with H3K4me3 and trimethylation of lysine residue 27 on H3 (repressive marker H3K27me3), have been shown to designate genes poised for activation [52–54]. In the case of intergenic regions, the H3K27ac marker is characteristically thought to distinguish active enhancers from their inactive or poised counterparts [9]. Our experimental results imply that this classic model does not faithfully apply in select physiological circumstances. Uncovering the time point–specific enrichment of inactive or poised enhancer-associated histone markers, such as histone H3 monomethylation of lysine residue 4 (H3K4me1) and H3K27me3 [9,11], would pinpoint more precisely which intergenic regions display an enhancer signature and when such regions become active. Supplemented with our current established H3K27ac and H3K4me3 enrichment profiles, a further tracing of promoters' and enhancers' gain, maintenance, or loss of other histone marks in a time course–dependent manner can offer a more developed view of the progressive changes in chromatin state at key contractility-driving genes.

Though presenting an inaugural view of the myometrial epigenetic landscape, the already validated pronounced enrichment of two prominent activation-associated histone marks at labor-associated gene loci as early as day 15 does itself suggest the following: that such a

molecular setup may explain the ease with which labor can occur in advance of term, if portions of the genome that are critical for contractility onset already contain accessible chromatin. This notion presents interesting implications for opposing critical conceptions of preterm birth, which characterize the event either as a pathological deviant version of its healthy default norm or as an evolutionary adaptive response that can act as a survival mechanism for the mother and/or fetus [55]. From the former perspective, epigenetic activation of chromatin in myometrial cells at the start of the late gestation period allows for early locus access to transcriptional machinery, but a prematurely timed stimulus prior to the end of term detrimentally activates the contraction-associated expression program too early, resulting in delivery of the fetus before term. As for the latter viewpoint, early access to genomic regions within labor-associated gene loci allows the dam to give birth earlier if such timing provides some kind of fitness advantage. For instance, the fetus' exit from unfavourable in utero conditions may be enabled via preterm delivery to promote its survival, at the risk of subjecting the underdeveloped pup to potentially intolerable ex utero conditions. Alternatively, some have argued that the dam may induce an early birth as a pregnancy termination strategy if fetal complications are not deemed worth the cost of the especial maternal energy investment in the late gestational period [55]. In either speculative case, the epigenetic priming event we have observed offers a novel mechanistic angle from which to view the potential initiation of preterm birth.

Finally, the molecular basis of labor can be better understood with a further examination of the activity of transcription factors driving regulation of contractility-driving genes. To date, a partial profile of the transcription factors that may bind key regulatory regions has been put forward. During pregnancy, JUN proteins are known to be present even at early gestational stages [23], prior to the expression of FOS proteins. Indeed, our RNA-seq data accordingly show that levels of Jun and Jund mRNA are similar across both gestational time points, whereas transcript levels of Fos, Fosb, and Fosl2 increase from day 15 to the onset of labor (S1 Table). Interactions between JUN transcription factors and corepressor proteins in quiescent tissues have been previously demonstrated [56]. The potential binding of JUN homodimers at AP-1 motifs within labor-associated gene promoters may explain why these genes are not activated prior to term. As term approaches, however, FOS subfamily proteins are up-regulated in response to hormonal signals and mechanical stretch stimuli [23,57]. These events result in an accumulation of FOS:JUN heterodimers, which, we suggest, may bind the same AP-1 motifs in gene promoters, but consequently exert an activating rather than repressive effect on labor–up-regulated genes at this time. Furthermore, the observed enrichment of the RELA motif at these promoters is unsurprising given prior studies highlighting the protein's role in the prolabor inflammatory pathway. An increased abundance of phosphorylated RELA immediately prior to the onset of labor [58] suggests that this protein may be a prominent player in the laboring transcription factor network. The other two enriched motifs we found for proteins affiliated with the promotion of gene activation—TCF3 and CTCF—also implicate them in the potential regulation of labor-associated promoter activity. TCF3 may perform a similar coactivator-assisting role that it has been shown to perform in other tissues [36]. CTCF has been proposed to anchor the interactions between gene promoters and distal regulatory elements because of its enrichment at both regulatory regions across cell types [37], a function that this protein may well also enact in the myometrium. Such molecular contributions, as well as the identities of all transcription factor interaction partners for the AP-1 factors controlling labor onset, are yet to be determined.

In this study, we have established a general picture of active chromatin states within the quiescent and contractile myometrial genome and the transcriptional events accompanying the establishment of such states. We have also provided evidentiary support for a broader regulatory role for AP-1 and RELA proteins in regulating these changes across multiple genomic

regions. Establishing a more fine-tuned understanding of the molecular basis of birth can allow for a more comprehensive list of therapeutic targets for the prevention of preterm labor in women.

# Materials and methods

## Ethics statement

This study was carried out in accordance with standards set out by the Canadian Council on Animal Care (CCAC). Animal Use Protocol AUP# 21-0164H was reviewed and approved by the Animal Care Committee of The Center for Phenogenomics (TCP), Toronto, Canada. All animals in this AUP were maintained and used in accordance with the current recommendations of CCAC, the requirements under the Animals for Research Act, RSO 1990, and The Centre for Phenogenomics Committee Policies and Guidelines. All research using animal tissues was performed in a class II certified laboratory by qualified staff trained in biological and chemical safety protocols and in accordance with Health Canada guidelines and regulations.

## Animal model

Bl6 or C57/Bl6 mice used in these experiments were purchased from Harlan Laboratories (http://www.harlan.com/). All mice were housed under specific pathogen–free conditions on a 12L:12D cycle and were administered food and water ad libitum. Female mice were mated overnight with males, and the day on which vaginal plugs were detected was designated as day 1 of gestation. Pregnant mice were maintained until the appropriate gestational time point. The day of delivery was day 19 of gestation. Our criteria for labor were based on delivery of at least one pup.

## Tissue collection

Animals were euthanized by carbon dioxide inhalation, and myometrial samples were collected on gestational day 15, day 19 term not in labor term-not-in-labor, day 19 during active labor, and 2–6 hours postpartum. Tissue was collected at 10 AM on all days with the exception of the labor sample (during active labor), which was collected once the animals had delivered at least one pup. The part of the uterine horn close to the cervix from which the fetus was already expelled was removed and discarded; the remainder was collected for analysis. For each day of gestation, tissue was collected from 4–6 different animals. Uteri were placed into ice-cold PBS. Uterine horns were bisected longitudinally and dissected away from both pups and placentas. The decidua basalis was cut away from the myometrial tissue. The decidua parietalis was carefully removed from the myometrial tissue by mechanical scraping on ice, which removed the entire luminal and glandular epithelium and the majority of the uterine stroma. Myometrial tissues were flash-frozen in liquid nitrogen and stored at −80˚C. When necessary, myometrial tissues were crushed into fine powder on dry ice prior to subjection to the experimental methods listed below.

## Chromatin immunoprecipitation

Histone marker–targeting chromatin immunoprecipitation (ChIP) was conducted using the protocol described by Young laboratory (younglab.wi.mit.edu/hESRegulation/Young_Protocol.doc), with some necessary modifications for myometrial tissue. Crushed myometrial tissue was fixed in a 1% paraformaldehyde solution at room temperature, and the reaction quenched in a 0.125 M glycine solution. Cells were rinsed twice with 1× cold PBS, and pellets were flash-frozen and stored at −80˚C until needed. Pellets were washed and samples were lysed in

successive lysis buffers, followed by subjection to sonication via the Covaris sonicator with a pulse ON time of 10 seconds at 30 amps for a total of 30 cycles. Aliquots of sonicated sample were run on a 2% agarose gel to confirm chromatin was sonicated to a range of 300–500 bp in size. Sonicated samples were treated with 10% Triton X-100 and spun down at 4˚C to pellet debris. Aliquots of cell lysate supernatant to be used as input were stored at −20˚C.

To bind antibody to magnetic beads, Dynal Protein A and Protein G beads (added in a 1:1 ratio) were washed and resuspended in block solution. Anti-H3K27ac antibody (Abcam, ab4729) or anti-H3K4me3 (Abcam, ab8580) was added, as appropriate, to beads and incubated at 4˚C with rotation. Beads were again washed and resuspended in block solution. Antibody and magnetic bead mix was added to remaining cell lysate, and samples were incubated at 4˚C overnight with rotation. IP samples were washed at 4˚C and eluted at 65˚C. Supernatant was removed from spun-down beads, and cross-links were reversed at 65˚C. Samples were treated with RNaseA at 37˚C, treated with Proteinase K at 55˚C, cleaned via phenol-chloroform treatment, and stored in ethanol at −20˚C overnight. DNA pellets were washed with 80% EtOH and resuspended in Tris-HCl. Validation of the ChIP method was performed using ChIP-qPCR primers (sequences in S11 Table) that targeted regions expected to be enriched in our marker of interest.

RNAPII-targeting ChIP was performed as outlined in the supplemental methods in Mitchell and Fraser [59], with modifications for collection of myometrial tissue as performed in the case of the histone ChIP, and using anti-RNAPII (RPB1 serine 5 phosphorylated form) antibody (Abcam, ab5131).

## ChIP sequencing and mapping

ChIP ($n = 2$) and input ($n = 1$) samples from each gestational time point—day 15, day 19 term-not-in-labor, day 19 term labor, and postpartum—were submitted for single-end 50-bp read sequencing using standard Illumina HiSeq 2500 protocols. Reads were quality checked using FastQC, trimmed with bbduk, and mapped to the GRCm38/mm10 mouse reference genome using STAR [60].

## ChIP normalization and peak calling

Peaks were called for each individual replicate using MACS2 broad peak calling [61]. Significantly conserved peaks in both biological replicates were combined using irreproducible discovery rate (IDR). Differential peak analysis was performed using the diffBind package [62]. Peaks with a fold change $\geq 4$ and adjusted $p$-value $< 0.01$ were considered significantly different between day 15 and term labor samples. Peaks with a significant increase in signal intensity in labor samples were linked to the closest gene TSSs using bedtools [63]. Normalized ChIP-seq reads (RPM) at promoters (+/− 2 kb of TSSs) of labor-associated up-regulated genes in H3K4me3-, H3K27ac-, and RNAPII-targeted samples were quantified using Seqmonk (https://www.bioinformatics.babraham.ac.uk/projects/seqmonk/). Kruskal–Wallis test was used to measure significant ($p < 0.05$) changes in enrichment values (RPM) among different time points. NGS.plot [64] was used to plot the K-means clustering heatmaps using default settings. Results were plotted using ggplot2 [65]. Sequencing data files were submitted to the Gene Expression Omnibus (GEO) repository (GSE124295).

## Gene expression quantification by RNA extraction and RT-qPCR

Total RNA was extracted from crushed myometrial tissue using Trizol and further treated with DNaseI to remove genomic DNA. RNA was reverse transcribed using the high-capacity cDNA synthesis kit (Thermo Fisher Scientific). Target gene expression was monitored by

qPCR using exon–intron boundary-spanning primers (S12 Table) for primary transcript detection and normalized to levels of total Hist1 mRNA because the *Hist1* reference gene was consistently expressed at similar levels across gestational time points. Expression levels were calculated against Bl6 or F1 genomic DNA-based standard curve references. All samples were confirmed not to have DNA contamination because no amplification was observed in reverse transcriptase negative samples. Relative expression values were plotted using GraphPad Prism 8. Significant changes in expression were determined by one-way ANOVA with Tukey correction.

## RNA-seq quantification and differential expression analysis

DNAseI-treated total RNA samples isolated from day 15 and term labor mice ($n = 5$ each) were subjected to paired-end sequencing using standard Illumina HiSeq 2500 protocols. Reads were quality checked using FastQC, trimmed with bbduk, and mapped to the GRCm38/mm10 mouse reference genome using STAR [60]. Exon-mapped reads were quantified using feature-Counts [66]. Intron reads were quantified using SeqMonk's active transcription quantitation pipeline (http://www.bioinformatics.babraham.ac.uk/projects/seqmonk/). Alternative transcript counts were summed together for every gene. Intron reads were then imported into DESeq2 [67] for differential expression analysis. Genes with a fold change $\geq 4$ and adjusted *p*-value $< 0.01$ were considered significantly changing. Differential RNA expression data were plotted using the EnhancedVolcano package (https://github.com/kevinblighe/EnhancedVolcano). Reads were normalized for gene expression levels across replicates. Heatmaps were plotted using the pheatmap package (https://cran.r-project.org/web/packages/pheatmap/index.html). Genomic regions of interest for eRNA expression analysis were selected based on intergenic regions featuring H3K27ac peaks. RNA-seq data at these regions were subjected to differential RNA expression analysis by DESeq2. H3K27ac peaks with an eRNA fold change $\geq 4$ and adjusted *p*-value $< 0.05$ were considered as peaks with significantly changing signal intensity from day 15 to day 19 (labor), and differential eRNA expression was plotted using EnhancedVolcano package. Sequencing data files were submitted to the GEO repository (GSE124295).

## Motif enrichment analyses

Enrichment of transcription factor motifs at promoters and intergenic regions of labor-associated genes was performed using HOMER motif analysis tool [35]. Promoter sequences (−1 kb of TSSs) of up-regulated genes at labor were compared against an input of randomly selected 1-kb-length promoter sequences. Intergenic regions containing labor-associated H3K27ac peaks and exhibiting significant up-regulation of eRNA expression were compared against random input sequences of varying size. Significantly enriched motifs in the HOMER database were calculated with *p*-value $< 0.05$.

## Metagene analyses

Genome-wide exon-normalized counts were divided into four quartiles according to the average expression of genes (transcripts per million [TPM]) across replicates in either day 15 or term labor time points. Expression quartiles were used to plot the average H3K4me3 and H3K27ac signal and RNAPII coverage at each individual time point using ngs.plot [64]. Significantly up-regulated and down-regulated intron-corresponding reads in term labor samples were used to plot H3K4me3 and H3K27ac signal and RNAPII coverage at all time points using ngs.plot.

### GO term analyses

A list of genes meeting our differential expression cutoff in the exon-based RNA-seq analyses was used as input in the EnrichR database. Outputs denoting genes' associated molecular functions, cellular components, biological processes, and pathways were sorted according to *p*-value.

## Supporting information

**S1 Fig. Gestational time point–specific RNA-seq samples cluster based on gestational time point of sample collection.** Hierarchical clustering of RNA-seq samples from d15 and d19 when in active labor. Darker color indicates increased correlation. Data associated with this figure can be found in S1 Data. d, day; RNA-seq, RNA-sequencing.
(PDF)

**S2 Fig. Proof of ChIP selectivity in myometrial tissues.** Applied anti-H3K27ac ChIP in murine myometrial tissue (target, pink) and murine embryonic stem cells (cell control, blue) revealed enrichment or lack of enrichment at select gene targets, as expected. Cell-specific enrichment of this histone mark observed at gene promoters expected to be active predominantly in myometrium rather than embryonic stem cells (left), in both cell types (center), and in neither cell type (right). Data associated with this figure can be found in S4 Data. ChIP, chromatin immunoprecipitation; H3K27ac, H3 acetylation on lysine residue 27.
(PDF)

**S3 Fig. Enrichment of activating histone marks at gene promoters depends on transcriptional status of genes.** Metagene plots displaying H3K4me3 or H3K27ac enrichment +/− 2 kb of TSSs for genes in expression quartiles reveals increased modification at the promoters of highly expressed genes. Data associated with this figure can be found in S5 Data. H3K4me3, H3 trimethylation of lysine residue 4; H3K27ac, H3 acetylation on lysine residue 27; TSSs, transcription start site.
(PDF)

**S4 Fig. Epigenetic landscapes of select labor-associated genes.** UCSC genome browser views of epigenetic and transcription regulatory mark enrichment profiles at (A) *Fosl2*, (B) *Gja1*, (C) *Oxtr*, and (D) *Ptgs2*. Data displayed include H3K4me3 signal (green), H3K27ac signal (blue), and RNAPII signal (burgundy) at d15, TNIL, LAB, and pp stages. Reads are mapped to mm10 assembly. d, day; *Fosl2*, Fos-like antigen 2; *Gja1*, Gap junction alpha 1; H3K4me3, H3 trimethylation of lysine residue 4; H3K27ac, H3 acetylation on lysine residue 27; LAB, active labor; pp, postpartum; *Oxtr*, Oxytocin receptor; *Ptgs2*, Prostaglandin G/H Synthase 2; RNAPII, RNA polymerase II; TNIL, term-not-in-labor; UCSC, University of California Santa Cruz.
(PDF)

**S5 Fig. H3K27ac and H3K4me3 enrichment levels at promoters in the myometrial genome exhibit no gestational age–specific enrichment trends.** Heatmaps representing promoters clustered according to H3K27ac and H3K4me3 enrichment levels, as per K-means clustering analysis. d15, TNIL, LAB, and pp time points indicated accordingly. H3K4me3, H3 trimethylation of lysine residue 4; H3K27ac, H3 acetylation on lysine residue 27; LAB, active labor; pp, postpartum; TNIL, term-not-in-labor.
(PDF)

**S6 Fig. Gestational time point–specific RNAPII ChIP-seq samples cluster based on gestational time point of sample collection.** Hierarchical clustering of RNAPII ChIP-seq samples from d15 and d19 when in active labor. Darker color indicates increased correlation. Data

associated with this figure can be found in S9 Data. ChIP-seq, chromatin immunoprecipitation with massively parallel sequencing; d, day; RNAPII, RNA polymerase II.
(PDF)

**S7 Fig. Enrichment of RNAPII at gene bodies depends on transcriptional status of genes.** Metagene plots displaying RNAPII enrichment at genes in expression quartiles reveals increased association at the promoters and gene bodies of highly expressed genes. Data associated with this figure can be found in S5 Data. RNAPII, RNA polymerase II.
(PDF)

**S8 Fig. Motif enrichment at promoters of labor–up-regulated genes.** Motif enrichment at promoters (1 kb upstream of TSSs) of genes with increased expression in labor based on intron reads. Data associated with this figure can be found in S6 Data. TSSs, transcription start site.
(PDF)

**S1 Data. Original numerical values for S1 Fig.**
(XLSX)

**S2 Data. Original numerical values for Fig 1C.**
(XLSX)

**S3 Data. Original numerical values for Fig 2D.**
(XLSX)

**S4 Data. Original numerical values for S2 Fig.**
(XLSX)

**S5 Data. Original numerical values for S3 and S7 Figs.**
(XLSX)

**S6 Data. Original numerical values for Figs 3C and 4A and S8.**
(XLSX)

**S7 Data. Original numerical values for H3K4me3 data in Fig 3D.** H3K4me3, H3 trimethylation of lysine residue 4.
(XLSX)

**S8 Data. Original numerical values for H3K27ac data in Fig 3D.** H3K27ac, H3 acetylation on lysine residue 27.
(XLSX)

**S9 Data. Original numerical values for S6 Fig.**
(XLSX)

**S10 Data. Original numerical values for Fig 4B.**
(XLSX)

**S1 Table. Genome-wide exon read-based RNA expression values in d15 and term laboring myometrium.** d, day.
(XLSX)

**S2 Table. GO terms for labor–down-regulated genes.** GO, gene ontology.
(XLSX)

**S3 Table. GO terms for labor–up-regulated genes.** GO, gene ontology.
(XLSX)

**S4 Table. Genome-wide intron read-based RNA expression values in d15 and term laboring myometrium.** d, day.
(XLSX)

**S5 Table. Correlation of gestational time point replicates in H3K4me3 and H3K27ac targeted ChIP samples.** ChIP, chromatin immunoprecipitation; H3K4me3, H3 trimethylation of lysine residue 4; H3K27ac, H3 acetylation on lysine residue 27.
(PDF)

**S6 Table. Reads per million of H3K4me3 at gene promoters.** H3K4me3, H3 trimethylation of lysine residue 4.
(XLSX)

**S7 Table. Reads per million of H3K27ac at gene promoters.** H3K27ac, H3 acetylation on lysine residue 27.
(XLSX)

**S8 Table. Genome-wide called broad RNAPII peaks in d15 and d19 laboring myometrium.** d, day; RNAPII, RNA polymerase II.
(XLSX)

**S9 Table. Reads per million of RNAPII at gene bodies.** RNAPII, RNA polymerase II.
(XLSX)

**S10 Table. Genome-wide called broad H3K27ac and H3K4me3 peaks in d15, term-not-in-labor, labor, and postpartum myometrium.** d, day; H3K4me3, H3 trimethylation of lysine residue 4; H3K27ac, H3 acetylation on lysine residue 27.
(XLSX)

**S11 Table. H3K27ac peaks displaying a significant increase in read counts in laboring samples relative to d15 samples.** d, day; H3K27ac, H3 acetylation on lysine residue 27.
(XLSX)

**S12 Table. Genome-wide eRNA expression values in d15 and term laboring myometrium.** d, day; eRNA, enhancer RNA.
(XLSX)

**S13 Table. List of primers used in ChIP-qPCR test.** ChIP-qPCR, ChIP-quantitative polymerase chain reaction.
(PDF)

**S14 Table. List of primers used in RT-qPCR experiments.** RT-qPCR, reverse transcriptase-quantitative polymerase chain reaction.
(PDF)

## Author Contributions

**Conceptualization:** Jennifer A. Mitchell.

**Data curation:** Virlana M. Shchuka, Luis E. Abatti.

**Formal analysis:** Virlana M. Shchuka, Luis E. Abatti, Huayun Hou, Nawrah Khader.

**Funding acquisition:** Virlana M. Shchuka, Michael D. Wilson, Oksana Shynlova, Jennifer A. Mitchell.

**Investigation:** Virlana M. Shchuka, Luis E. Abatti, Nawrah Khader.

**Methodology:** Luis E. Abatti, Anna Dorogin, Michael D. Wilson, Oksana Shynlova, Jennifer A. Mitchell.

**Project administration:** Virlana M. Shchuka, Oksana Shynlova, Jennifer A. Mitchell.

**Resources:** Oksana Shynlova, Jennifer A. Mitchell.

**Software:** Luis E. Abatti, Huayun Hou.

**Supervision:** Jennifer A. Mitchell.

**Validation:** Virlana M. Shchuka, Luis E. Abatti, Nawrah Khader.

**Visualization:** Virlana M. Shchuka, Luis E. Abatti, Jennifer A. Mitchell.

**Writing – original draft:** Virlana M. Shchuka, Luis E. Abatti, Jennifer A. Mitchell.

**Writing – review & editing:** Virlana M. Shchuka, Luis E. Abatti, Huayun Hou, Nawrah Khader, Anna Dorogin, Michael D. Wilson, Oksana Shynlova, Jennifer A. Mitchell.

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
