## [Editor Report · Decision Letter 0]

8 Mar 2020

***HAVE YOU ASSIDear Dr Mitchell, 

Thank you for submitting your manuscript entitled "The pregnant myometrium is epigenetically activated at contractility-driving gene loci prior to the onset of labor in mice" for consideration as a Research Article by PLOS Biology.

Your manuscript has now been evaluated by the PLOS Biology editorial staff as well as by an academic editor with relevant expertise and I am writing to let you know that we would like to send your submission out for external peer review.

Please re-submit your manuscript within two working days, i.e. by Mar 10 2020 11:59PM.

Kind regards,

Di Jiang,

Associate Editor

PLOS Biology

---

## [Decision Letter · Decision Letter 1]

6 Apr 2020

Dear Dr Mitchell,

Thank you very much for submitting your manuscript "The pregnant myometrium is epigenetically activated at contractility-driving gene loci prior to the onset of labor in mice" for consideration as a Research Article at PLOS Biology. Your manuscript has been evaluated by the PLOS Biology editors, an Academic Editor with relevant expertise, and by three independent reviewers.

In light of the reviews (below), we will welcome re-submission of a much-revised version that takes into account the reviewers' comments. Regarding reviewer 2's point 1 suggesting looking at an earlier gestational time point, we'd like you to include the data if already available, and you will need to address this at least in the discussion. We also wish to emphasise the need to separate in the revised text very clearly what conclusions are based on the data and what statements are speculation as suggested by the reviewers. We cannot make any decision about publication until we have seen the revised manuscript and your response to the reviewers' comments. Your revised manuscript is also likely to be sent for further evaluation by the reviewers.

We expect to receive your revised manuscript within 2 months. 

**IMPORTANT - SUBMITTING YOUR REVISION**

*Re-submission Checklist*

*Published Peer Review*

*PLOS Data Policy*

*Blot and Gel Data Policy*

Sincerely,

Di Jiang, PhD

Associate Editor

PLOS Biology

REVIEWS:

Reviewer's Responses to Questions

Reviewer #1: The manuscript by Shchuka and colleagues assessed whether or not the pregnant mouse myometrium was epigenetically activated at contractility-driving gene loci just prior to labour. This appears to be a revised manuscript so it is not known what was revised since a marked version was not provided, but it was very well presented. The work is excellently written with a solid introduction, excellent rationale and objective. The figures were well laid out, well described and appropriately interpreted. Overall, the manuscript significantly advances our understanding of the epigenetic and transcriptional alterations that occur in uterine smooth muscle prior to parturition. I only have one minor comment to provide. 

Minor Comment

1. In the Discussion, lines 334-336, the authors indicated that the results explain the effortlessness with which labour an occur before term. The authors should take the opportunity to speculate a little bit as to why that would be an evolutionary advantage or advantage otherwise. Would this allow the pregnant dam to abort their fetus(es) in the event of a stressful, dangerous or other situation since the DNA is open and accessible and just needs a specific event(s) to set in motion labour? Could this be a survival strategy for the dam in a way?

Reviewer #2: In this manuscript, the authors performed simultaneous RNA-seq and ChIP-seq analysis of pregnant (days (d)15, d19-not-in-labor [NIL], d19-in labor [LAB]) and post-partum C57Bl/6 mouse myometrium to assess changes in gene expression in association with genome-wide binding of the active histone marks, H3K27ac and H3K4me3, and of RNA-polymerase II (RNAPII, S5-p) to promoters and distal enhancers. They identified 533 genes (including known contraction-associated genes) that were upregulated at term, compared to d15. Surprisingly, the promoters and potential intergenic enhancers of these genes manifested enrichment of the active H3K27ac and H3K4me3 histone marks as early as d15, and binding of these histone marks remained relatively unchanged at d19-NIL and d19-LAB. By contrast, binding of RNAPII to the promoters and potential distal enhancers of contraction-associated genes was markedly increased at d19-LAB, compared to d15. Based on these findings, the authors conclude that 'epigenetic activation of the myometrial genome precedes active labor by at least 4 days in the mouse, suggesting that these genes are poised for rapid transcriptional activation at term.' Since the promoter and enhancer regions are enriched in AP-1 motifs, and since Fos is one of the labor-associated genes in myometrium, an elaborate model is presented, indicating Jun binding in both quiescent and laboring myometrium, and enhanced Fos and RNAPII recruitment to both promoters and distal enhancers during labor.

 The studies are well executed and the manuscript is clearly written; however, the epigenetic mechanisms that lead to the increase in RNAPII binding and transcriptional activation of the labor-associated genes at term remain undefined. While an interesting hypothesis is proposed, a number of essential components are lacking from this study to support this. These include the following:

Major points:

1. The assumption that chromatin is 'open' in myometrium as early as d15 is based on the finding that binding of the active histone marks, H3K27ac and H3K4me3 don't change between d15 and d19-LAB. However, to what is d15 being compared? It would be important to compare binding of these histone marks to an earlier gestational time point (i.e. is binding at d15 significantly increased compared to d5 or d10?), and to non-pregnant myometrium. Rather than repeating the ChIP-seq study, the authors could select promoter regions of several known contraction-associated genes and use ChIP-qPCR to analyze binding of these modified histones. 

2. No evidence is presented that chromatin is actually 'open' at d15. Certainly, the two histone marks chosen (H3K27ac and H3K4me3) may not tell the whole story. Binding of the active histone marks H3K9ac, H3K14ac, as well as the repressive marks, H3K9me2/3 and H3K27me3, should be analyzed, as further indices of active and inactive chromatin. This again could be performed using ChIP-qPCR of selected contraction-associated genes at a range of gestational time points. Binding of endogenous selected histone deacetylases (HDACs) also may provide important insight regarding the regulation of dynamic changes in chromatin accessibility. Additional time points between d15 and d19-LAB should be analyzed, since one would expect activating epigenetic changes to occur prior to increased RNAPII binding and increased gene transcription 

3. The hypothesis set forth in the text and in Figure 5 suggests that Jun is bound during myometrial quiescence and that increased Fos expression and promoter/enhancer binding increases near term, resulting in enhanced RNAPII binding and transcriptional activation of contractile genes. However, this is conjecture. Temporal changes in the binding of endogenous Jun and Fos and their association with active/inactive histone marks and RNAPII binding should be included to provide critical insight into the mechanisms underlying transcriptional activation of labor-associated genes in the pregnant myometrium. 

Additional comments:

4. Lines 165-167 and Figure 2D - This reviewer finds it very surprising that expression of the contraction associated genes were only significantly increased in laboring myometrium (d19-LAB), compared to d15. According to published studies, they should be increased by d19-NIL, but this does not appear to be the case.

5. Figures 3 and 4 - The manuscript is highly focused on epigenetic changes within and surrounding the Fos gene. However, additional genome browser views of other differentially regulated contraction-associated genes should be included to compare gestational changes in binding of modified histones and RNAPII to support their conclusions.

Reviewer #3: The manuscript "The pregnant myometrium is epigenetically activated at contractility-driving gene loci prior to the onset of labor in mice" by Shchuka et al investigates the chromatin states of the quiescent and contractile myometrial genome and correlates the changes with transcriptional events that result in the establishment of such states. Specifically, the authors identified a set of contractility-associated genes that are highly upregulated at active labor (E19) compared to pre-labour (E15) due to higher transcription. Using genome-wide approaches to explore histone modifications, the promoters associated with these genes show a very high level of active histone modifications such as H3K27ac and H3K4me3. The main novel finding in this manuscript is that the level of H3K27ac was enriched at all time points (prelabour, labour and post-labor), suggesting that the transcription of labor-associated is not regulated by chromatin changes active histone marks (ie the promoters are already in an active state). The authors further show that the transcription of labor-associated genes correlated to high enrichment of RNPII, suggesting that although the promoters contain active histone marks, something else is recruiting RNAPII to drive transcription. Thus, the mechanism driving transcription at labour is more complex. The authors also identified enhancers that have H3K27Ac in pre-labour and at labour, but also display increased enhancer RNA production and RNAPII only at labour. Using motif analysis of the promoters and enhancers, the authors suggest that AP-1 and RELA may play a role in driving transcription of genes at labour onset. 

The manuscript is very well written and presents novel findings. The idea that labour genes have active promoters and enhancers and that an additional level of regulation controls transcriptional onset is extremely interesting. The quality of the data is high, including replicates as appropriate. However, I am less convinced that AP-1/RELA is the mechanism involved. I have provided comments for the author outlined below.

Comments: 

1. The authors present the model that JUN family protein plays a role in maintaining myometrial gene expression during pregnancy but heterodimerization with FOS is required for regulation of labor associated genes. This model is very interesting. However, the authors fail to show this experimentally. If possible, data examining FOX/JUN and/or RELA binding would greatly strengthen the model. Motif analysis alone does not determine if binding is actually occurring. Thus, Figure 5, the model figure presented in the manuscript is not fully experimentally validated. 

2. For the motif analysis, the authors identified motifs enriched in the promoters of labour onset genes. However, for this analysis they used random sequences for background. Would it be more appropriate to use all promoters or promoters of genes that are unchanged to show enrichment?

3. Histone modification such as H3K4me1 has been used to identify enhancer and its overlap with H3K27ac determines the active state of the enhancer. Why did the authors not investigate the level of H3K4me1 to identify all enhancer and investigated their active, poised and inactive states at the different time points? This may help strengthen the enhancer part of the model by identifying enhancers in pre-labour. Please clarify.

4. On page 11, the authors indicate that they are looking at "other intergenic regions of interest". That is too vague and should be clarified.

5. It is not always clear for the histone mod analyses whether the authors are looking at all promoters, or only the promoters of the labour-onset genes. This should be clarified throughout the manuscript.

6. Fig 1B, several genes are downregulated upon onset of labor. What are those genes and what are their cellular/molecular functions? The author could perform GO analysis of differentially expressed genes at these time points 

7. Figure 1B and 2B; add the number of genes in the figure. 

8. Figure 3B, why did the authors not perform K-means clustering to identify regions with different H3K27ac/H3K4me1 enrichments at d15, TNIL, d19 LAB, pp?

9. Figure 4C; add H3K27ac, RNPII and eRNA signals as to the FOS genomic maps. 

10. Consider adding motif analysis results in the main figures. 

11. What are the mRNA levels of FOS/JUN/AP1 at different time-points? RNA-seq data can show their mRNA levels. 

12. Figure 5, similar to the labouring stage, the author should draw regulatory regions (enhancers) in the quiescent stage.

---

## [Editor Report · Decision Letter 2]

27 May 2020

Dear Dr Mitchell,

Thank you for submitting your revised Research Article entitled "The pregnant myometrium is epigenetically activated at contractility-driving gene loci prior to the onset of labor in mice" for publication in PLOS Biology. I have now obtained advice from the Academic Editor who has evaluated your revisions and the response to the reviewers. 

Based on the evaluation, we're delighted to let you know that we're now editorially satisfied with your manuscript. However before we can formally accept your paper and consider it "in press", we also need to ensure that your article conforms to our guidelines. A member of our team will be in touch shortly with a set of requests. As we can't proceed until these requirements are met, your swift response will help prevent delays to publication. Please also make sure to address the data and other policy-related requests noted at the end of this email.

*Copyediting*

*Published Peer Review History*

*Early Version*

*Submitting Your Revision*

Sincerely,

Di Jiang, PhD

PLOS Biology

ETHICS STATEMENT:

-- Please create a separate subsection entitled "Ethics Statement" and place it in the beginning of the Methods section. 

-- Please include the full name of the IACUC/ethics committee that reviewed and approved the animal care and use protocol/permit/project license. Please also include an approval number.

-- Please include the specific national or international regulations/guidelines to which your animal care and use protocol adhered. Please note that institutional or accreditation organization guidelines (such as AAALAC) do not meet this requirement.

-- Please include information about the form of consent (written/oral) given for research involving human participants. All research involving human participants must have been approved by the authors' Institutional Review Board (IRB) or an equivalent committee, and all clinical investigation must have been conducted according to the principles expressed in the Declaration of Helsinki.

DATA POLICY:

-- Regardless of the method selected, please ensure that you provide the individual numerical values that underlie the summary data displayed in the following figure panels as they are essential for readers to assess your analysis and to reproduce it: Figures 1BC, 2BD, 3ACD, 4ABDE, S1, S2, S3, S6, S7, S8. NOTE: the numerical data provided should include all replicates AND the way in which the plotted mean and errors were derived (it should not present only the mean/average values).

-- Please provide a reviewer/editor key/token for the sequencing data files deposited in Gene Expression Omnibus (GEO; https://www.ncbi.nlm.nih.gov/geo/) repository (GSE124295) so we can check it before accepting the paper. 

---

## [Editor Report · Decision Letter 3]

19 Jun 2020

Dear Dr Mitchell,

On behalf of my colleagues and the Academic Editor, Rocio Melissa Rivera, I am pleased to inform you that we will be delighted to publish your Research Article in PLOS Biology. 

Early Version

PRESS 

Kind regards,

Alice Musson

Publishing Editor, 

PLOS Biology

on behalf of

Di Jiang, PhD,

Senior Editor

PLOS Biology